# The Freeze-Drying of Foods—The Characteristic of the Process Course and the Effect of Its Parameters on the Physical Properties of Food Materials

**DOI:** 10.3390/foods9101488

**Published:** 2020-10-18

**Authors:** Dorota Nowak, Ewa Jakubczyk

**Affiliations:** Department of Food Engineering and Process Management, Warsaw University of Life Sciences, Nowoursynowska 159C, 02-776 Warsaw, Poland; dorota_nowak@sggw.edu.pl

**Keywords:** freeze-drying, shelf temperature, lyophilization pressure, physical properties, foods, sublimation, desorption

## Abstract

Freeze-drying, also known as lyophilization, is a process in which water in the form of ice under low pressure is removed from a material by sublimation. This process has found many applications for the production of high quality food and pharmaceuticals. The main steps of the freeze-drying process, such as the freezing of the product and primary and secondary drying, are described in this paper. The problems and mechanisms of each step of the freeze-drying process are also analyzed. The methods necessary for the selection of the primary and secondary end processes are characterized. The review contains a description of the effects of process conditions and the selected physical properties of freeze-dried materials, such as structural properties (shrinkage and density porosity), color, and texture. The study shows that little attention is given to the mechanical properties and texture of freeze-dried materials obtained from different conditions of the lyophilization process.

## 1. Introduction

Freeze-drying is a process in which water is sublimated by the direct transition of water from solid (ice) to vapor, thus omitting the liquid state, and then desorbing water from the “dry” layer [1,2,3,4,5]. It is widely used for the stabilization of high-quality food, biological materials, and pharmaceuticals, such as proteins, vaccines, bacteria, and mammal cells. In the process, the quality of the dried product (biological, nutritional, and organoleptic properties) is retained [6,7]. This is due to the fact that freezing water in the material prior to lyophilization inhibits chemical, biochemical, and microbiological processes. Therefore, the taste, smell, and content of various nutrients do not change. Raw food materials contain a lot of water, ranging from 80% to 95%. The removal of water by sublimation results in the creation of highly porous structure of the freeze-dried products, and the rehydration of lyophilisates occurs immediately [8,9].

The water in the products can be free water or water bound to the matrix by various forces. Free water freezes, but bound water does not freeze. In the freeze-drying process, all ice water and some bound water must be removed. Therefore, lyophilization is a highly complex and multi-step process that consists of [1,6]:The freezing of the product, most often under atmospheric pressure.Primary drying—proper freeze-drying—ice sublimation, most often at reduced pressure.Secondary drying—desorption drying—drying the product to the required final humidity.

The effect of freeze-drying should be considered from the economic aspect and the quality of the freeze-dried material. The cost of a product mainly depends on the freeze-drying time. Therefore, process parameters and other conditions of its course are often set so that its time is as short as possible. Setting parameters to speed up the process can lead to the deterioration of the product’s properties. For example, increasing shelf temperature can lead to the defrosting of the product and the collapse of the structure or to the thermal degradation of heat-sensitive food ingredients.

The conditions of the freeze-drying process should be selected in a way that does not melt the water. Liquid water is the reaction medium and changes the rheological properties of the product. The presence of liquid water during the freeze-drying of food products may result in many changes in the composition, morphology, and physical properties of foods (e.g., shrinkage). It may also reduce the period of ensuring high quality during storage [10]. The color and structure–texture properties are crucial in the quality evaluation of food by consumers. Therefore, the dependence of these food properties on the parameters of freeze-drying is extremely important.

The effect of freeze-drying conditions on the nutritional properties, antioxidant activities [11,12,13,14], and glass transition characteristics [12,15,16] of different food materials can be found in the literature. It is widely believed that freeze-drying is the best method of drying. However, improperly selected process parameters may cause unfavorable changes in the material, such as shrinkage, color change, collapsed structure. Therefore, the aim of this review was to characterize all stages of the freeze-drying process, discuss the phenomena taking place during those stages, present their impact on the course of the process, and explain the effect of the process conditions on the selected physical properties of different food products.

## 2. The Characteristics of the Freeze-Drying Process

During the three stages of the freeze-drying process (sublimation, primary drying, and secondary drying), six main physical phenomena can be distinguished that have a significant impact on the course of the process, the quality of the obtained material, and the overall costs of the process. Those are:The phase transition of the water contained in the product into ice.The ice to vapor phase transition.The desorption of water molecules from material structures.The obtainment of a sufficiently low pressure.The re-sublimation of water vapor removed from the material on the surface of the condenser.The removal of a layer of ice from the surface of the capacitor.

Both the kinetics of the process and the properties of the obtained product depend on the parameters in which these phenomena occur.

The main feature of freeze-drying, the only one that distinguishes it from vacuum drying, is the need to keep free water frozen. This is one of the most difficult problems in freeze-drying.

Freeze-drying is a mass exchange process that requires heat transport. The heat of sublimation is 2885 kJ/kg [17,18]. If too little heat is supplied, the process will be slow, which will increase its costs. If the supplied heat flux is too high, it will cause an accumulation of heat in the material and an increase in its temperature, consequently leading to the possibility of the appearance of liquid water. Hence, it is extremely important to maintain a balance between the amount of heat supplied and used. One way to assess whether the amount of heat supplied is too high is to monitor the temperature of the lyophilized material [19]. Its value must not exceed the value of the cryoscopic temperature for a given material or the glass transition temperature for a given water content. If the glass transition temperature is exceeded, the structure may collapse (porosity reduction), which is highly disadvantageous due to the reduction of the specific surface of the product. As a consequence, the time of the second drying stage lengthens, the rehydration capacity of the product deteriorates, the product has a higher final water content [20]. Moreover, it may result in lower product stability during storage.

Maintaining a constant, low temperature (according to the pressure in the chamber) during the sublimation period proves that the balance between the amount of heat supplied and used for sublimation is maintained. However, it does not mean that the process runs at the maximum possible sublimation rate under given conditions. Too low a value of the supplied heat flux may limit the sublimation rate. On the other hand, an increase in temperature may indicate that the heat input is too high. It can be also the effect of the possible heat consumption by sublimation due to increase of the heat transport resistance. Therefore, for more complete control, changes in the water content should also be monitored simultaneously [19].

The parameter that determines the amount of heat supplied is the heat transfer resistance, while the resistance to mass movement (water vapor), both inside and outside the material, determines the amount of heat used for evaporation. Therefore, the course of lyophilization is determined by all the factors affecting the value of both these resistances. These factors are related to the process parameters at each stage, the design of the freeze-dryer, and the properties of the lyophilized material.

### 2.1. The First Stage of the Freeze-Drying Process—Freezing the Raw Material

Though freezing is one of the most critical stages during lyophilization, the importance of the freezing process has been rather neglected in the past [4].

Both solid products (fruits and vegetables) and liquid products (coffee and juices) must be frozen before freeze-drying [4,21,22]. Freezing is the crystallization of a solvent that, in the case of food products, is water. The additional benefits resulting from the conversion of water into ice in the material before the freeze-drying are as follows:Immobilizing the ingredients in solution and preventing foaming occurring during pressure reduction in the freeze-dryer chamber.Limiting the chemical, biochemical, and microbiological changes taking place in the material.Creating a specific structure of ice crystals in the frozen product, which, in the next step, facilitates or limits the migration of water vapor from the dried material; the structure of ice formed during freezing determines the intensity of mass movement and, as a result, shapes the final morphology of the dried material [23].Stiffening of the structure, counteracting contraction of the cells of plant or animal tissue caused by the removal of water from them, which is possible due to the plasticization of the material by liquid water.

When the cryoscopic temperature is exceeded, the water begins to crystallize and the solution or cell juice cryo-concentrate, which leads to the transport of the dry substance components. Thus, freezing should be carried out to avoid concentration gradients within the frozen tissue. For this reason, it is advantageous to obtain a state of supercooling (lowering the material temperature below the cryoscopic temperature), which causes it to occur simultaneously in the entire volume of the material and accelerates the freezing process [22,24].

The freezing rate is very important for freeze-drying; the kinetics of ice nucleation and crystal growth determine the physical state and morphology of the frozen cake (layer of frozen material) and, as a result, the properties of the freeze-dried product [25]. Ice morphology is directly correlated with the rate of sublimation both in primary and secondary drying.

Taking the course of lyophilization into account, the appropriate freezing rate depends on the type of material—whether it is a solution, suspension, or biological material with a cell structure. To optimize the freeze-processes of liquid materials, the ice crystal sizes must be large enough to obtain the shortest primary drying time. The formation of numerous, small ice crystals during the freezing step leads to a high resistance to mass transfer in the dried product, whereas the formation of a few large ice crystals leads to small resistance [7]. However, to intensify the secondary drying period, these sizes need to be smaller to provide a large specific surface area for the dried matrix. As a result, the desorption of non-frozen water from pores on the surface of the amorphous matrix proceeds faster [7]. Therefore, the method of freezing should be decided after analyzing the course of primary and secondary drying. If the secondary drying takes a long time, it can be accelerated by changing the freezing conditions to obtain a lot of small ice crystals. Ceballos et al. [26] investigated the effect of freezing rate on the properties of freeze-dried soursop fruit pulp. After six hours of lyophilization in the obtained material, a higher water content was correlated with a higher freezing rate. This proved a higher resistance to mass movement in the case of a quickly frozen material. For a biological material with a cell structure in which one wants to preserve the biological functions of the cell membrane, quick freezing should be used so that ice crystals do not damage it. Likewise, the rate of freezing may influence the behavior of bioactivity by protein molecules. The cell membrane resists mass transport; therefore, its destruction may facilitate the diffusion of water vapor. Thus, slow freezing may be preferable. Similarly, when freeze-dried solutions or other materials lacking an internal porous structure are freeze-dried, large crystals, after sublimation, leave spaces to facilitate sublimation.

Cellular materials have a certain porosity that positively influences mass transport resistance [27]. In the case of frozen liquids, it is possible to create a porous structure of the material and reduce the heat and mass transport path via the application of the new spray-freezing into liquid (SFL) technology [28,29,30,31,32,33]. By spraying the liquid into liquid nitrogen or cold air, it is possible to obtain a material in the form of ice spheres and thus obtain a porous bed that is then freeze-dried. (Figure 1). The result is a granulate with the desired particle diameter. In this method, it is possible to obtain a very high freezing rate due to the small particle diameter (low resistance to heat transport) and the large contact surface with the cooling medium. The freezing time is in the millisecond range. The sublimation rate is also high due to the low resistance to heat and mass transport as the effects of the small particle diameter. This technique allows for a high degree of control over the residual moisture content, mass density, and particle size. It allows for the easy manipulation of process parameters such as the temperature of the cryogenic liquid, the chemical composition and concentration of the solution, and the atomizer type. It allows for the obtainment of liquid powders with the desired particle diameter [30]. A certain disadvantage of the method is the limitation of heat conduction due to the small contact surface of the frozen product spheres with the heating surface and the empty spaces between the spheres. Therefore, radiant heating from the top shelf is important, while the positive effect of voids is a low resistance to water vapor movement [33].

The freezing rate is directly proportional to the temperature difference between the cryoscopic temperature of the material and the temperature of the freezing medium, and it is inversely proportional to the heat transfer resistance. The temperature difference is a value that can be easily adjusted, especially when freezing in outdoor units [19].

The heat transport resistance for a given material depends on the thickness of the layer. The smaller the thickness, the lower the resistance and the faster the process. However, the smaller the thickness of the material, the less material is processed. Spin freezing is a way to reconcile these two mutually exclusive aspects. During spin freezing, the product is frozen before freeze-drying in unit packages. The desired amount of material, resting on the bottom of the package, forms a fairly thick layer. Swirling the package during freezing creates a layer of frozen material, with a much smaller thickness and a larger evaporation surface, on the walls (Figure 2). This method allows you to regulate the freezing rate in a much larger range of values [35]. The oxidizable liquid or semi-liquid food products are freeze-dried. It is worth recommending this method for the lyophilization of this type of product in unit packages similar to vials. It significantly reduces the drying time. In addition, it gives the possibility of vacuum packaging in the freeze-dryer chamber, which allows one to obtain a product protected against oxidation.

A way to speed up freezing, especially in the case of large-sized materials where the conduction resistance prevents rapid freezing, is to use supercooling. In the absence of crystal seeds, the material must be supercooled below the cryoscopic temperature for nucleation to form. The greater the degree of subcooling, the faster the freezing process takes place in the entire volume of the material. A higher degree of supercooling increases the rate of ice nucleation and the effective rate of freezing, yielding a high number of small ice crystals [35,36]. In a study by Searles et al. [36], it was found that the primary drying rate is about 4% lower for each degree of additional supercooling.

The supercooling state can be achieved by various methods, e.g., by adding cryoprotectants [37], thanks to the external magnetic field that prevents the movement of water molecules to the crystal surface [38,39]. Another way is freezing at elevated pressure [40]. By taking advantage of the pressure dependence of the freezing temperature, the product can be subjected to super atmospheric pressure and then reduced in pressure, causing it to freeze.

If it is possible to evaporate from the free surface, the self-freezing effect can be used. If the water begins to intensively evaporate, it takes away heat from the product from which it evaporates, causing it to quickly freeze in its entire volume without the need for separate unit operation [41].

Silva et al. [42] studied the freezing process of coffee extract in comparison with air-freezing and contact; vacuum freezing led to significantly smaller freezing times. At the same time, some of the solvents were evaporated, thanks to which additional compaction of 26–43%, contrasting with 1–2% losses for air freezing and contact freezing, was obtained. This fact, along with the extremely porous structure formed during vacuum freezing, makes this method particularly interesting for soluble coffee production by freeze-drying. Such a phenomenon may occur in the initial stage of drying—during the reduction of pressure in the freeze-dryer chamber. However, there are restrictions. In the case of a non-cellular material, there is intensive evaporation during pressure reduction, and since the viscosity of the liquid at low temperature is high, it causes the material to splash. In the case of a cellular material that provides a high mass transport resistance, there are two cases for the use of this phenomenon. Evaporation can be facilitated when the material is naturally thin (e.g., vegetables or other deciduous plants) or in the case of a thin scrap of tissue with damaged cell membranes by cutting [7]. Another way to reduce mass transport resistance is by destroying cell membranes. Such an effect can be obtained by acting on the structure with a pulsed electric field (PEF). PEF pretreatment provokes the damage of cell membranes and accelerates mass and heat transfer processes without undesirable changes in food tissues [43]. Parniakov et al. [44] showed that the reduction of the temperature of an apple slice treated with PEF depends on the degree of structure destruction. With the degree of cell wall disintegration equal to 0.96, while reducing the pressure in the freeze-dryer to a value of 1000 Pa, the temperature of the apple decreased from 25 to −10 °C due to evaporation; meanwhile, for undamaged tissue, the temperature only decreased to approximately −5 °C.

### 2.2. Primary Drying—Sublimation

Primary drying is the term for the freeze-drying period in which the ice sublimation process takes place. When designing the lyophilization process for a given material, the following process parameters should be determined: the pressure in the freeze-dryer chamber and the intensity of heat supply. The flux of the supplied heat depends on the heating method. In the case of contact heating, setting the appropriate shelf temperature is necessary. When the material is heated by radiation, the distance from the material, the range of infrared radiation, and the intensity of radiation should be selected [45]. In the case of microwave heating, the intensity of the microwaves and the duration of their operation are important. Another process parameter is the temperature of the condenser surface, but it is a parameter that results from the design of the cooling cycle in the freeze-dryer. The temperature of the condenser surface should be from about −60 to −80 °C [46], depending on the type of freeze-dried material. Therefore, the set pressure in the chamber, the vapor pressure above the sublimation surface, and the condenser temperature should be determined at the design stage of the refrigerant circuit. It should be remembered that the desublimation or deposition surface temperature increases during the process. The ice layer that forms is the heat transfer resistance that increases with increasing ice thickness. This resistance causes a temperature gradient between the ice surface and the capacitor surface. It is therefore important to remove this layer during the process, or a large capacitor surface area should be designed [46] so that the ice layer is not too thick.

During the sublimation period, the amount of heat supplied should correspond to the amount of heat necessary for the sublimation of the ice. Heat can be supplied by heat conduction [47,48,49], heat radiation [6,45,50,51,52], or microwave heating [53,54,55,56,57,58,59,60,61]. For the sublimation process to proceed, two basic conditions must be met: the sublime steam must be constantly removed from the sublimation area, and to maintain the differential pressure of the vapor resulting in the removal of water vapor from the chamber, the heat necessary for sublimation must be continuously supplied to the material. If either of these two basic conditions are not met, some unfavorable phenomena occur, such as the softening, thawing, bulging, or collapse of the structure [46].

The sublimation process takes place from the surface of the product during pressure reduction in the freeze-dryer. During lyophilization, the ice–vapor phase boundary moves into the material [19,62] or, in the case of porous and heterogeneous materials, from places where the resistance to mass movement is the lowest. The driving force of the sublimation process is the difference between the vapor pressure above the sublimation surface *p*_iw_ (as the saturation pressure corresponding to the temperature *T*_i_ of the sublimation surface) and above the evaporator surface *p*_a_ (as the saturation pressure corresponding to the surface temperature *T*_a_ on which the resublimation process takes place). To initiate sublimation, the pressure in the chamber must be significantly lower than the vapor pressure of the ice resublimated on the condenser [4].

The total resistance to mass movement consists of three components: resistance to movement inside the dry layer *R*_d_, resistance to mass transport from the surface of the dry layer to the resublimation surface *R*_s_, and ice sublimation resistance *R*_l_.

Thus, the sublimation rate can be described by the formula [63]:(1)$$G=\frac{p_{iw}-p_{a}}{R_{d}+R_{s}+R_{l}}$$
where *G* is the ice sublimation rate, kg/(m^2^∙s), *R*_d_ is the resistance inside the dry layer; m^2^/(Pa∙s kg), *R*_s_ is the resistance to mass movement from the dry surface to the resublimation surface; m^2^/(Pa∙s kg), and *R*_l_ is the ice sublimation resistance; m^2^/(Pa∙s kg).

The ice sublimation resistance *R*_l_ can be expressed by the formula:(2)$$R_{l}=\frac{\sqrt{T_{i}}}{K_{l}}$$
where *T*_i_ is the temperature corresponding to the saturation pressure and *K*_l_ is a constant, depending on the molecular weight of the sublimating substance; for water, *K*_l_ = 0.018.

Assuming pure ice sublimation (no dry layer resistance), the *T*_i_ temperature corresponds to the saturation pressure *p*_iw_. The evaporator temperature *T*_a_, corresponding to the saturation pressure *p*_a_, is much lower than that of the *p*_iw_ (*p*_a_ << *p*_iw_); the sublimation surface is also the surface of the ice, so *R*_d_ = 0. It can also be assumed that the resistance of the convective mass transport from the evaporation surface to the resublimation surface is negligible. Under such conditions, the sublimation rate reaches the maximum value *G*_max_ [62]:(3)$$G_{max}=\frac{p_{iw}}{R_{l}}=\frac{p_{iw}K_{l}}{\sqrt{T_{i}}}$$

Assuming that the ice sublimes at −10 °C, i.e., the saturation pressure is 260 Pa, and then: *G*_max_ = 0.29 kg/(m^2^∙s). If the sublimation temperature of ice drops to −20 °C, *G*_max_ will decrease by half, and at the sublimation temperature of −30 °C, *G*_max_ will reach the value of 0.04 kg/(m^2^∙s). Thus, by lowering the working pressure in the sublimation chamber, the ability to sublimate is significantly reduced [59]. In practice, this value will be lower due to the resistance to movement of the mass created by the dry layer. The dried product resistance to mass transfer (*R*_p_) depends on several parameters, namely (1) the rate of freezing, related to the formation of ice crystals with different sizes and morphologies; (2) the formulation, related to the quantitative and qualitative dry substance content and total layer thickness; and (3) the processing conditions during primary drying, related to the possible formation of micro-collapse [7].

Assegehegn et al. [46] tested the sublimation rate in vials using pressures from 5 to 20 Pa and at shelf temperatures in the range from −10 to −30 °C. At a pressure of 15 Pa and a shelf temperature of −20 °C, they achieved a sublimation rate of 0.17 g/h; at a pressure of 5 Pa and the same temperature, it was 13 g/h for the vials placed in the front of the freeze-dryer and 0.11 and 0.8 g/h in the center of the freeze-dryer. These results also indicated that the sublimation effect differs depending on the position inside the freeze-dryer chamber. Therefore, when determining the pressure of the freeze-drying process, its value should not be reduced too much without substantive reasons. The higher its value, the greater the temperature difference between the vapor generated and the condenser (the higher the temperature, the faster the water vapor is removed from the chamber). The limitation is the chemical composition of the sublimated material. The pressure in the freeze-dryer chamber must be low enough for the free water in the product to solidify. This is especially important for materials with high contents of monosaccharides and salts, the presence of which lowers the cryoscopic temperature, and for materials containing cryoprotectants. The pressure in the freeze-dryer chamber should be below 610 Pa (the approximate triple point pressure of water). Usually, in the case of plant materials, it is around 63–124 Pa at a temperature from −25 to −20 °C and lower [64]. In the pharmaceutical industry (solutions of active proteins, sugars), this range is 5–20 Pa [46].

For sublimation to take place, the heat of phase change, which is 2885 kJ/kg, must be provided. The higher the driving force of the heat exchange process, the higher the heat flux, and, therefore, the higher the temperature difference between the material and the heat source. This difference is a tool for the process operator to control the heat movement process.

As reported in the literature [65,66], there are different mechanisms of heat transfer during primary drying. Therefore, the overall heat transport effect is related to direct conduction through the ice layer. The rate of heat transfer through direct conduction can be defined as the flux of heat conducted through the frozen layer of the material [62]:(4)$$q=\frac{\lambda_{i}}{x_{f}}\left(T_{lw}-T_{i}\right)$$
where *λ*_i_ is the ice specific thermal conductivity, W/(m∙K); *x*_f_ is the ice layer thickness, m; *T*_lw_ is the temperature of the lower ice layer, K; and *T*_i_ is the temperature on the border of sublimation, K.

Referring to Formula (4), if that heat is conducted through a 1 cm thick layer of frozen water, and the temperature difference between the inside of the material and its surface is 10 K, then a maximum of 2.24 kW/m^2^ is delivered to the evaporation surface. This flux is equivalent to the sublimation capacity of 7.9 × 10^−4^ kg/(m^2^ × s) of ice. This value is much lower compared to the *G*_max_ value. Thus, it can be concluded that the rate-limiting factor for sublimation is heat transport through the ice layer.

Assegehegn et al. [46], examining the dependence of the sublimation rate on the shelf temperature, found a proportional relationship between the shelf temperature and the sublimation rate. At a pressure of 5 Pa, an increase in temperature from −20 to −5 °C caused a two-fold increase in the sublimation rate, while in the experiment conducted at a pressure of 15 Pa, an increase in the shelf temperature from −30 to −17 °C caused an increase in the sublimation rate from 0.08 to 0.20 g/h. These results confirmed that heat transport through the ice layer is the rate-limiting factor for sublimation. If adequate energy is not supplied during the sublimation period, the product temperature drops until the appropriate vapor pressure and pressure in the chamber reach dynamic equilibrium. At this point, no net ice sublimation is observed [46].

If the process parameters are not selected properly, the structure may collapse (porosity reduction), which is highly unfavorable due to the reduction of the specific surface of the product. As a consequence, the time of the second drying stage lengthens, the rehydration capacity of the product deteriorates, and the product has a higher final water content and a poor visual assessment [20]. Moreover, it may result in a lower product stability during storage.

It was found that the resistance to mass movement imposed by the dry layer depends not only on the freezing conditions but also on the process parameters. For the same conditions of pressure, shape, layer thickness, and location in the freeze-dryer chamber, the dried product resistance decreased from 1.94 to 1.37 mbar h/g with an increase in shelf temperature from −30 to −17 °C [46].

### 2.3. Second Drying—Desorption

The desorption drying process, also known as post-drying, takes place under reduced pressure with the simultaneous heating of the product to the assumed water content, determined individually for each raw material. At this stage, the drying rate is significantly reduced compared to the sublimation process due to the small amount of water, high resistance to heat, and mass transport through the porous material layer, as well as the bonding of water particles by the components of the dry substance, especially those constituting a monolayer [67].

It is generally accepted that water exists in solid matrices in three different forms that correspond to specific regions of the sorption isotherm [68]. In the first region corresponding to water activity values below 0.2, water molecules form a monolayer, are tightly bound to the solid matrix through hydrogen bonds, and are inaccessible to the reaction. In the second region, corresponding to water activities between 0.2 and 0.5, the water is loosely bound, forming a multilayer. In this area, water molecules no longer form hydrogen bonds with the components of the dry matter. The molecular interactions of water–water dominate, which favors the formation of microscopic regions of condensed water. This form of water may already constitute a solvent and reaction medium. The third form of water corresponds to a water activity greater than 0.5. It is relatively free water, fills the capillaries, and complies with Raoults’ law [69] in primary drying. The freeze-drying process removes water that mainly corresponds to the first two forms of water. Free water crystallizes in the freezing process and is removed in the first phase of drying [70]. The duration of secondary drying can make up a significant proportion of the process time, especially if the moisture content of the final product is low. The final moisture content of the lyophilizate is a critical parameter, as it determines the stability and = storage stability of the product. The final equilibrium humidity depends on the parameters of desorption drying. Too high or too low a value of this moisture content is unfavorable; too high is not favorable for long-term storage, too low may damage the active material. It is assumed that the final equilibrium dry matter content should be higher than 95% [62]. For pharmaceutical products, target dry matter contents as high as 98–99% or higher are common. At such high dry matter contents, the water contained in the monolayer is removed [67].

Depending on the chemical composition, structure and freezing conditions, the components of the dry substance may be in an amorphous state (always when the freezing rate is high) or in a crystalline state. In the case of a large number of amorphous components, the water content after sublimation drying is complete can potentially affect storage stability. Its content may be 5–20% (depending on the solids content in the preparation) of the initial water content [67]. On the other hand, the water content in crystalline materials after sublimation is complete is insignificant because all available water crystallizes during the freezing stage [67].

The dry substance matrix can also absorb water that is transported from the sublimation surface through the “dry” layer. During the desorption drying process, the heating temperature of the material must be lower than the maximum temperature allowable for the material due to the possibility of thermal degradation and the possibility of a transition to the rubber state. Thus, the maximum allowable temperature results from the specifics of a given material and is determined individually. In the case of protein drugs, the maximum allowable temperature should be lower than 40 °C [67,71,72], and for food products, fruits, and vegetables, the maximum allowable temperature may reach 60–70 °C or higher but always below the glass transition temperature.

The fulfilment of this condition is very important because it affects storage stability. The physicochemical properties of the glass state (molecular mobility, viscosity, viscosity changes, structural breakdown, crystallization, etc.) change with changing water content. When the vitreous material is stored below its *T*_g_, the reaction rate is lower at a low water content because the diffusion and mobility of the reactants are limited. As the water content increases, the *T*_g_ of the formulation is lowered to a temperature below the storage temperature. Then, the material becomes rubbery and the mobility of the dry substance constituent molecules increases, the stability of the product to decrease. Therefore, in this state, the bioactivity of the ingredients can significantly change. For this reason, secondary drying significantly determines the characteristics of the freeze-dried material [68,69,73].

Regardless of the reasons, it is very important to remove any remaining water after freeze-drying is complete. The lower the water content, the higher the glass transition temperature of the dried product (*T*_g_), which significantly affects its storage stability. The dried product should be stored at a temperature well below its *T*_g_ to avoid structure collapse and flow. Residual water acts as a plasticizer and increases the mobility of dry matter particles, facilitating various types of unfavorable transformations. The rate of desorption is strongly dependent on the temperature. Trelea [70] found that in the usual temperature range encountered in freeze-drying, desorption is more than three times faster at 40 than at 10 °C.

During secondary drying, adsorbed (non-frozen) water is desorbed [73]. Liapis and Bruttini [74] found that there are several mechanisms of water removal in secondary drying: These are (1) simultaneous adsorption and desorption at the interface between the surface of pores and gas, (2) convective transport in pores, (3) gas diffusion in pores, (4) the diffusion of water in the solid particles, and (5) the diffusion of water on the surface of the solid. They concluded that the first three mechanisms were rate-limiting during desorption. The pressure in the chamber is usually kept at the same level as in the first period or slightly lower. However, to increase the desorption rate, the chamber pressure should be made as low as possible.

Millman et al. [75] conducted research on the development of an optimization model for secondary drying conditions (chamber pressure and shelf temperature) based on the assumed final product’s moisture content and the maximum allowable product temperature limits. They found that the shortest time necessary to achieve the assumed moisture content primarily depends on the selected end criterion of the process, including the average or maximum moisture content in the dry layer. In this study, a different water content was noted after a certain amount of time, depending on the position of the sample in the freeze-dryer chamber. The model developed by Sadikoglu et al. [76,77] minimized the process duration due to the variability of the shelf temperature and chamber pressure profiles over time within the maximum allowable product temperature limit at critical locations and the final product moisture requirements. Both of these models assume their one-dimensional movement from the top to the bottom of the sublimation boundary and the distribution of temperature and moisture in the porous layer. Subsequent studies, taking the two-way movement of heat and mass into account, confirmed the necessity to modify the desorption parameters during its duration [78,79]. An additional benefit of this procedure is that there was less variation in moisture contents in samples taken at different locations in the chamber. The study of Sadikoglu et al. [78] also provided information on the number and location of samples that should be monitored in real-time to ensure product stability and quality.

Recent research on the modelling of secondary drying has been based on the use of a dynamic desorption model as a software sensor for monitoring secondary drying. It links the model with the measurement of the desorption rate, which allows for the determination of the residual moisture content of the product and the kinetic parameter of desorption in real-time. From there, it is possible to estimate, in real-time, the remaining amount of water at the end of the first drying, the change in product moisture during secondary drying, and the time remaining to reach the target moisture content [80,81]. Trelea [70] introduced an additional aspect to the model assumptions. Additionally, they considered significantly different desorption kinetics for water molecules with different degrees of association with the solid matrix, as well as the dependence of the desorption rate on temperature. As a result, a model of the equilibrium moisture and desorption kinetics of the lyophilized preparation of lactic acid bacteria was developed.

### 2.4. Methods of Determining the End of Primary and Secondary Drying

The separation mechanisms of mass and heat transport during primary and secondary drying, as well as other methods of process intensification, indicate the necessity to identify the moment of transition from primary drying to secondary drying. During primary drying, some of the material is a dry layer and some is solidified water. Thus, there is no clear boundary between the first and the second phases of freeze-drying, and both processes can occur simultaneously for a certain time [76,82]. Taking the fact that primary drying is the sublimation of the solidified solvent into account, the end of primary drying should be considered to be the moment when the last ice crystal sublimates and only the bound water remains in the material. By monitoring the freeze-drying process, it is possible to find changes in the intensity of increase or decrease in the value of parameters characteristic for this process, such as material temperature, humidity, or water vapor pressure in the freeze-dryer chamber [74]. The course of the kinetics of the loss of material mass is also changing. These dependencies are used in the methods of determining the end of primary drying: the comparative pressure measurement of Pirani, the capacitance manometer, the pressure rise test, product temperature or shelf surface temperature response, and the kinetics of mass changes [20].

If there is water vapor in the chamber of the freeze-dryer, the values of the Pirani gauge—operating on the thermal conductivity principle—are higher than the readings of the capacitive gauge, which shows absolute pressure regardless of the gas composition. The point where the Pirani pressure begins to rapidly drop is the end of sublimation [20].

Another way to determine the end of primary drying is the pressure increase test, which relies on checking whether sublimated water vapor from the material is still accumulating in the freeze-dryer chamber. This can be checked by cutting off the condensation chamber for a short time. When sublimation is not complete, the pressure in the chamber increases [20].

The temperature of the product remains relatively constant during the primary drying phase so long as there is ice sublimation, followed by an increase of temperature [20], but it can also mean the overheating of the material and the melting of ice. Therefore, it is also worth controlling on-line changes in the water content (based on mass changes) and recognizing primary drying as completed when the water content corresponds to the multilayer capacity, determined based on sorption isotherms for a given product.

An example of this control method is shown in Figure 3. The values of water contents, obtained after the time after which the material temperature increased to the cryoscopic temperature both at the surface and along the material axis, are presented on the drying rate curve. These values are also the values at which the sublimation process is finished. If these values are much higher than the water content corresponding to multilayer capacity for a given material, then the ice melted (Figure 3A).

Reaching the cryoscopic temperature when the lyophilized material has a water content that is higher than the bound water content indicates the melting of the ice (Figure 3A); meanwhile, when it is reached with a sufficiently low water content, the process is correct (Figure 3B).

When the weight of the material does not change within 30 min, it can be assumed that the equilibrium water content, under process condition, has been reached.

## 3. Effect of Freeze-Drying Conditions of the Selected Physical Properties of Materials

The physicochemical and structural changes of food products during processing may significantly affect the final product quality. Color changes and a lack of color stability are important problems that occur during food treatment and storage. Additionally, processing procedures have an impact on porosity, texture, taste, the retention of nutrients, and the sorption of materials [15]. The application of the drying process leads to many changes in the physical, chemical, and nutritional properties of foods [83]. Freeze-drying is a less damaging process than air-drying and spray-drying [84]. For this reason, lyophilization is recognized as the best food dehydration method [85]. Controlling the freezing rate, temperature level, total gas pressure, and final mean moisture content is required to obtain a freeze-dried product with adequate quality [84].

### 3.1. The Shelf Temperature

The process parameters of freeze-drying may significantly affect many quality attributes of foods and other materials subjected to this kind of dehydration. The temperature of the heating plate is one of the parameters that plays an important role in creation of the material structure during the freeze-drying. Undesired changes in the material structure can be a result of inappropriate temperature applied in the process [54]. An excessive collapse of the structure may lead to a decrease of the freeze-drying rate during secondary drying, as well as the deterioration of many features of products related to texture, porosity, volume, shape, stickiness, rehydration capacity, and stability [20,67]. Alves and Roos [86] showed that the appropriate conditions of the freeze-drying process have been identified by many researchers using a method of trial and error. Food materials are different in terms of structure, initial moisture content, and composition, and it is difficult to predict their behavior during freeze-drying. However, Antelo et al. [87] proposed a different approach based on a combination of dynamic modelling with the efficient and optimized off-line and on-line control of the freeze-drying process to obtain the required quality of products.

The mechanisms of drying and pore creation, as well the stability of products, are different during the primary and secondary stages of freeze-drying. The sublimation of ice is the longest part of the drying process, and it consumes more energy than during secondary drying stage [6,46,88]. For this reason, extensive work on designing the appropriate freeze-drying cycles, and especially primary drying for different food products, has been done [20,77,88,89].

Malik et al. [88] applied a different shelf temperature (−20, −30, and −40 °C) at the primary step of the freeze-drying of gum Arabic solutions with concentrations of 20–60%. The secondary drying was performed at 20 °C with a constant chamber pressure of 0.1 mbar. The investigation showed that the puffing effect was significant for the 60% concentration of Arabic gum in the sample when drying occurred at shelf temperatures of −30 and −20 °C. The porosity of freeze-dried hydrocolloid was also higher at the higher temperatures of the shelf during primary drying. Temperatures of −20 and −30 °C were not sufficient to cool down the sample because these temperatures exceeded the melting point.

The effects of the shelf heating temperature on the quality parameter and drying time of different freeze-dried products have been reported in the literature. Grapefruit puree was subjected to freeze-drying at room temperature without shelf heating and at a shelf temperature of 40 °C. It was observed that an increase of the shelf temperature caused a shortening of the drying time by more than 50% [16]. The traditional freeze-drying of mushroom (*Cordyceps militaris*) also showed a significant reduction of the drying time of about 37% when the process temperature was increased from 40 to 70 °C [13]. The sublimation rate is generally higher at higher temperatures, but material dried at excessive temperatures can collapse and lose the pore structure created by the freezing process [87]. In order to control product stability, especially during the secondary drying stage, the temperature of the product should be limited to between 10 and 35 °C for heat-sensitive materials, and for less heat-sensitive materials, the temperature can be higher than 50 °C [54,90]. Higher temperatures of freeze-drying at the secondary stage accelerate the drying process because more energy is required to remove the remaining water in the material [13,91]. However, too high a temperature may cause the melting of ice during the sublimation step of drying, resulting in structural changes such as shrinkage [54,92]. The selection of a proper shelf temperature should be based on the balance between the input and the required heat [13]. Additionally, the final quality of dried material is a crucial factor during the design of the freeze-drying process.

Egas-Astudillo et al. [16] reported that the application of higher temperatures during the freeze-drying of fruit puree enabled them to obtain a product with a higher quality and a lower process cost due to reduced drying time. Raising the shelf temperature from ambient to 40 °C promoted a slight increase in the porosity of freeze-dried grapefruit puree from 0.81 to 0.83. In addition, the mean area of pores also increased. The results obtained by Egas-Astudillo et al. [16] may indicate that not only was the structure of the material changed, but the mechanical resistance (fracture force) of the material dried at a lower heating plate temperature was also reduced.

The physical properties describe some features of food, which are particularly important to customers, such as the shape, volume, color, and texture of food products. The porosity of the freeze-dried product is a significant discriminant of its quality. The pores’ size and their distribution in the material have significant effects on the texture of foods, especially their crispness and crunchiness. Some works on the relationship between the shelf temperature, as well as the density, shrinkage, and porosity of freeze-dried products, have been presented in the literature [93,94]. Sabalni and Rahman [93] noted a decreasing trend of apparent porosity and an increase of apparent density with the increase of shelf temperature in the range from −45 to 15 °C for abalone, potato, and brown date. A similar tendency was also reported by Krokida et al. [95] for freeze-dried potatoes, carrots, and bananas. However, for freeze-dried-fruits, such as apples and yellow dates, the apparent porosity increased with an increase of the plate temperature. The shrinkage of the material was also lower [93]. The glass transition concept, which states that significant changes in the apparent porosity (reduction in pore formation) and the collapse of the structure occur at or close to T_g_, cannot be applied to all dried products [93,96]. The mechanism of pore formation and shrinkage during drying can be varied for foods because of their different structure, composition, initial porosity of materials [93,97], variety, ripeness, size, shape, pre-treatment prior to drying, and process conditions [98]. Some structures may also be more resistant to collapse during the drying process [93]. The investigations carried out by Sablani et al. [94] showed the applied shelf temperature of −5, −15, and −25 °C did not affect the final porosity and apparent density of freeze-dried garlic. However, the study showed that the porosity continuously increased with decreasing water content in the material during drying at a shelf temperature of −5 °C, whereas at lower shelf temperatures, the general tendency was similar but a fluctuation of porosity (drop) occurred when the critical moisture content was reached. At that stage, it was assumed that there was no ice remaining in the dried material, and the effect of temperature on pore formation was not observed.

The type of material and its composition may influence the shrinkage of the freeze-dried product. The shrinkage of material did not significantly differ under the different shelf temperatures of 20, 40, and 70 °C for freeze-dried apples (13%) and strawberry (8%) [15]. It was consistent with investigations of Shishehgarha et al. [99] for freeze-dried whole strawberries under various temperatures in the range from 30 to 70 °C. This study showed that the value of shrinkage was also independent of the shelf temperature but the number of collapsed fruits increased with the heating temperature. In the case of pear, a high shelf temperature (70 °C) maximally caused 12% shrinkage in the sample, but at lower temperatures, it resulted in a decrease of this parameter to around 6%. It was concluded that the susceptibility to shrinkage in the pear sample at high temperatures of freeze-drying could be related to the glass transition of the pear [15].

The porous structure of dried products may affect color descriptors, especially the lightness of material due to the presence of air voids and pores. The determination of the color parameters of dried products is crucial because the color of the product is one of the main quality criteria evaluated by consumers. Changes in the color of dried foods can be an indicator of the undesired thermal degradation of many bioactive compounds.

The effect of shelf temperature on the color of different products has been analyzed in some publications [11,85,100,101]. Krzykowski et al. [11] applied different shelf temperatures (20, 40, and 60 °C) during the freeze-drying of red pepper puree. The process time was shortened by more than half by raising shelf temperature in the range of 20–60 °C. A similar tendency was observed during the drying of whole and pulped cranberries when the temperature of the heating plate was increased from 30 to 70 °C. The time of drying was shorter by about 40% [100]. The freeze-drying of pepper puree increased its lightness and yellowness in comparison to fresh material. However, redness decreased at 20 °C but increased at 40 and 60 °C. Generally, the long drying time of pepper at 20 °C and the application of a high temperature of 60 °C led to a decrease of the color intensity due to the degradation of carotenoids [11]. The freeze-drying of cranberry pulp and whole fruits with increasing shelf temperature caused an increase of the lightness, redness, and color intensity. The intensive heat treatment caused changes in color as a result of the degradation of reddish anthocyanin pigments [100].

Hammami and Rene [101] noted that the lightness of freeze-dried skin and pulp did not change with the increase of the heating plate temperature. However, shelf temperatures higher than 60 °C caused a slight decrease in the lightness of strawberries, which can be linked to the presence of dark brown color at the surface of the fruit. The browning of the skin led to a significant decrease of color coordinates, which was caused by the excessive heating of products and non-browning or Maillard reactions. Similar phenomena were observed for freeze-dried apples when the heating shelf temperature exceeded 55 °C [85]. Silva-Espinoza et al. [14], during the freeze-drying of orange puree, did not observe changes in lightness L* with an increasing process temperature from 30 to 50 °C, but the chroma C* was enhanced at the higher shelf temperatures of 40 and 50 °C and the hue angle decreased for temperatures below 50 °C.

Khalloufi and Ratti [15] investigated the color changes of apple, pear, and strawberry during freeze-drying at shelf temperatures of 20, 40, and 70 °C. The color attributes did not differ significantly for apple and pear for lower heating temperatures of 20 and 40 °C. Additionally, in the case of strawberry, the increase of the *b** value in comparison to raw material was caused directly by the freeze-drying process, not by the intensity of heating. The higher temperature affected the browning of pear and apple. The glass transition temperature was used as an indicator of possible color changes. The highest value of *T*_g_ was observed for strawberry, whereas the lowest was observed for pear. It was concluded that strawberry should be less sensitive to color deterioration than the pear under the same freeze-drying conditions. Shishehgarha et al. [99] also reported that an increase of the heating shelf temperature to 70 °C did not have a significant effect on the color and volume of freeze-dried strawberry. During the freeze-drying of grapefruit puree, heating the material to 40 °C did not affect the color of the final product [16]. Martínez-Navarrete et al. [12] observed that freeze-drying of a mandarin snack preserved the orange color, which was characteristic for this type of fruit. Though the addition of biopolymers (Arabic gum, WPI-Whey Protein Isolate) affected the color of the snack more than increasing the shelf temperature to 40 °C.

Texture is one of the most important attributes in the quality evaluation of food products. Semi-empirical models have been used to describe the mechanical properties of porous foods based on density changes during processing [102,103,104,105]. Additionally, to predict the textural properties of foods based on their microstructure, finite element modelling has been applied [106,107,108]. Apart from mechanical properties, porosity and density are also used to characterize some texture features of dried products [109]. The state variables (temperature, moisture, and deformation during drying), the state of material (glassy and rubbery) and the external environment, as well as other drying conditions, may significantly affect the final texture of dried products [105]. Numerous publications have shown that the drying process influences the mechanical properties of different materials. Moist material subjected to drying is frequently viscoelastic, and with the progress of the dehydration process, the product becomes more brittle at a low moisture content [110,111,112,113].

The effect of shelf temperature on the texture of freeze-dried products can be found in a few research studies. Silva-Espinoza et al. [14] noted that the compressed cylinders of freeze-dried orange puree obtained at a higher shelf temperature behaved the same as a material with higher rigidity. Additionally, the values of the slope of the fracture curve were related to the sample resistance and the deformation increase with raising shelf temperature. The high temperature of 50 °C created the mechanical rigidity of the freeze-dried product before it fractured. It was an important feature of the dried material because it could be related to the higher mechanical resistance of the product during its transport.

Penetration tests were used to evaluate the texture of freeze-dried strawberries after rehydration. It was found that the temperature of the heating plate did not have a significant influence on the texture attributes of rehydrated strawberries [101]. A similar tendency was observed for freeze-dried apples [85]. The minimal texture degradation of freeze-dried strawberries was observed under the following optimal process conditions: a shelf temperature of 15 °C and a rate of temperature increase of 1.6 °C/min or a shelf temperature of 45 °C and a heating rate of 0.4 °C/min. The application of a low shelf temperature and a high rate of heating enabled them to maintain cell integrity. The high temperature of a shelf with a low heating rate prevented meltdown and cell collapse [114].

The puncture test was applied to measure some mechanical properties of the freeze-dried mandarin snack. The study showed that the heating of the shelves from room temperature to 40 °C had a significant effect on texture. The fracture force and area under the puncture curve decreased, but the number of peaks increased with an increase of the shelf temperature. The sample without biopolymer incorporation, freeze-dried at 40 °C, was estimated as a crunchy product with every fragile structure, but it was more stable during storage regarding mechanical parameters than the snack obtained at a lower heating temperature [12]. Mawilai et al. [115] investigated the effect of shelf temperature (30, 40, and 50 °C) on the texture of freeze-dried dragon fruits during secondary drying. The pulp dried at a temperature of 40 °C and was characterized by the lowest hardness and the highest crispness. However, hardness did not significantly differ for the dried peel in the case of an increase of temperature, but its crispness increased.

The textural properties of a dried material are strongly related to its water content and water activity [112]. Though these parameters are important for a proper evaluation of many other properties of freeze-dried foods, it has been noticed that many investigators do not always measure and control the values of the water content and water activity after freeze-drying. Table 1 shows the values of water activity and water content obtained for freeze-dried materials in different operational conditions. The water content for most of the freeze-dried products varied from 0.01 to 0.1 g/g d.m., and water activity ranged from 0.08 to 0.330.

### 3.2. The Pressure Chamber

The application of adequate working parameters of lyophilization, including the working pressure, may significantly affect the course of the process but also the quality of the obtained products [12,14,101,120]). The orange puree was freeze-dried at different pressures inside the chamber (5 and100 Pa). A significant effect of pressure on color attributes was observed. A low pressure led to higher values of lightness *L** and lower values of chroma *C**. Samples dried at a higher working pressure were darker and had a saturated color [14]. An increase of the working pressure from 12 to 100 Pa resulted in a decrease of in *L**, as well as an increase of the yellowness and greenness of the freeze-dried kiwi [121].

Udomkun et al. [122] carried out the freeze-drying of papaya slabs at three different levels of chamber pressure and at a constant shelf temperature of 20 °C. All color indexes were similar for samples dried at working pressures of 28 and 77 kPa. The application of a lower pressure of 9 kPa caused an increase of the *L**, *b**, and ΔE (total colour difference) values but a decrease of the *a** value. Hammami and Rene [101] also observed a decrease of the *L** value at a higher pressure for freeze-dried strawberries, and it was accompanied by a significant shrinkage of the sample. The applied pressure during freeze-drying should be lower than 50 Pa to reduce the shrinkage of sample fruits, such as strawberries.

The working pressure did not have an obvious effect on the apparent density, solid density and porosity of freeze-dried papaya. A slightly higher porosity was recorded for samples dried at the lowest pressure of 9 kPa. SEM micrographs showed that the pressure during freeze-drying had a noticeable effect on the changes in the surface and structure of the final product. The samples obtained at 28 kPa were characterized by a less collapsed and more homogeneous and porous structure than papaya samples dried at 9 and 77 kPa, the structure of which contained dense dry layers and large cavities. Oikonomopoulou, Krokida, and Karathanos [10] also observed a higher porosity and an increase of the number of large pores in the case of the application of lower working pressure for freeze-dried potato, mushroom, and strawberry. The bulk density of these freeze-dried materials decreased along with the decrease of the applied pressure (from 1.5 to 0.06 mbar).

Pressure has a strong impact on the sublimation process, especially the temperature of ice sublimation. The sublimation removal of ice creates pores and gaps with different characteristics. Therefore, the desired porosity, density, and final water content can be obtained by manipulating the freeze-drying parameters [10,95,123].

Structure descriptors, such as density and porosity, may characterize the texture of the dried products. Additionally, the formation of pores and their distribution in products during drying enables one to design the process and influences many different properties, such as the textural properties of foods [10,96,124].

Different operating pressures had a little effect on the fracture force of dried orange puree, although the effect was statistically significant. The authors supposed that a lower process pressure might result in the creation of a dried product with a more fracture-resistant texture [14]. On the contrary, Domin et al. [121] noted that a lower pressure caused a decrease of the penetration force. The higher force recorded for the kiwi samples obtained at a higher pressure was related to the collapse of the structure during freeze-drying. The process was carried out without the heating of the plate. Hence, the increase of pressure caused an increase of the freeze-drying temperature, which led to the collapse of samples.

### 3.3. The Freezing Rate

The freezing of materials before the sublimation process is also an important step in the creation of freeze-dried products. The freezing rate affects the structure of frozen and dried material.

The color attributes of freeze-dried orange puree, which was frozen at a slow rate (−45 °C in a conventional freezer) and a fast rate (−38 °C in a blast freezer), were characterized by similar mean values. The freezing rates also did not significantly affect texture [14]. Additionally, the slow and quick freezing did not influence the color indexes and texture of the freeze-dried carrot samples. However, the application of quick freezing in liquid nitrogen instead of slow freezing (a household freezer at −15 °C) reduced the shrinkage of freeze-dried carrot samples from 3.2% to 2.2% [125]. The effect of the freezing rate on the properties of freeze-dried products could be related to the initial structure and composition of materials. Genin and Rene [61] have reported that the texture, degree of ripeness, and moisture content in plant tissue have considerable impacts on the freeze-drying process. The loading density, the height of the product layer, the surface of the product, and condenser capacity may also affect the freeze-drying rate [101].

Samples of apples were frozen to −25 °C with different rates of 0.5, 2, and 3 °C/min, and then they were freeze-dried. Slow freezing reduced the drying time by 8.3% in comparison to the fastest freezing rate. The slow rate of freezing caused damage to the cell wall, which facilitated the moisture removal. The decrease of the cooling rate also resulted in a softer texture of apples and a higher rehydration capacity [126].

The structure of the freeze-dried maltodextrin–agar system showed that more heterogeneous pores were formed at freezing temperatures of −40 and −80 °C than at −20 °C [123]. The lower freezing temperatures and higher nucleation rates created homogeneous structures with more uniform and smaller pores [123,127,128]. The higher freezing temperatures resulted in less rapid nucleation and a longer time of ice crystal growth, which led to the occurrence of large pores. Maltodextrin–agar gel frozen at −80 °C and freeze-dried was most resistant to compression. It was stated that rapid freezing caused crust formation. The product also had smaller pores and thinner wall membranes. In addition, the gaps between walls were substantially limited. For this reason, the higher freezing rate caused an increased mechanical strength of the freeze-dried materials [123]. A similar effect of enhanced resistance to compression produced by a higher cooling rate was observed for freeze-dried alginate gels [129].

Table 2 summarizes the different conditions (shelf temperature, pressure chamber, and freezing temperature) applied during the freeze-drying of different products and their selected properties. 

## 4. Summary and Conclusions

The cited literature on the analysis of the phenomena occurring during the entire freeze-drying process comes mainly from the area of pharmacy. Maintaining the biological activity of this group of products, both after the process and after the storage period, is the overriding goal. Therefore, many studies concerning the influence of the individual stages of freeze-drying on the final activity of the product have been carried out. Therefore, a modern technologist should make every effort to preserve not only substances that are noticeable by the consumer but also vitamins and other bioactive substances, often very labile, that are degraded during processing. The presented publications also showed the effect of process conditions on the physical properties of freeze-dried foods. This has been analyzed by some authors, mainly concerning the shrinkage and porosity of dried material. The texture of a freeze-dried product obtained at different parameters of the process, such as the shelf temperature, chamber pressure, and the freezing rate, were investigated in a small range. Texture is a major feature assessed by consumers. This was surprising because the textural properties of food are strongly linked with the sensory analysis of food products. More publications can be found in regard to the color of the dried material. This is also important because changes in color during the freeze-drying process provide some information on the degradation of bioactive compounds such as antioxidants.

It should be understood what changes a selected material is subject to at each stage of freeze-drying, and the parameters of each stage should be selected based on the specificity of a raw material. The selection of proper conditions for the freeze-drying of a food material should be performed based on the characteristics of raw materials, such as composition (water content, the presence of sugars, proteins, and bioactive compounds), type of material (tissue, liquid material, semi-liquid material, and gel), and the glass transition temperature of foodstuff. The selection of freeze-drying parameters is sometimes arbitrary, but it should be based on preliminary research because each food material is different, and it is not possible to use some freeze-drying parameters for all types of food. The control of the heat supply is necessary to not exceed the melting point, which may lead to material degradation. Such control will ensure shorter drying time and the more uniform concentration of bound water at the end of secondary drying. This is a way of achieving a high-quality product.

## Figures and Tables

**Figure 1 foods-09-01488-f001:**
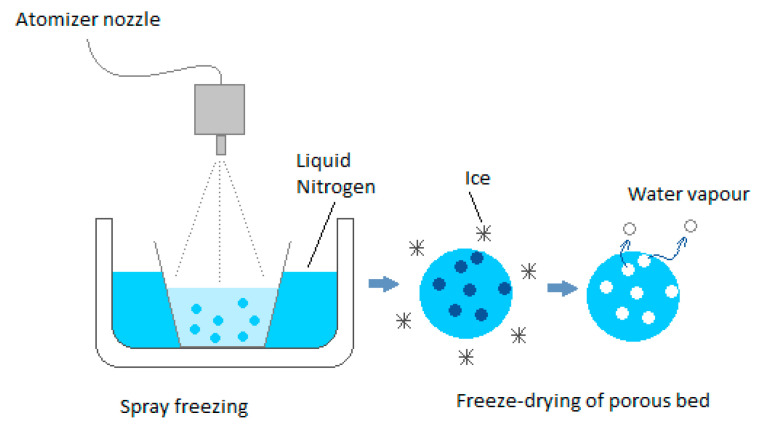
Producing a porous bed by freezing the suspension in a cryogenic fluid; adopted from [34].

**Figure 2 foods-09-01488-f002:**
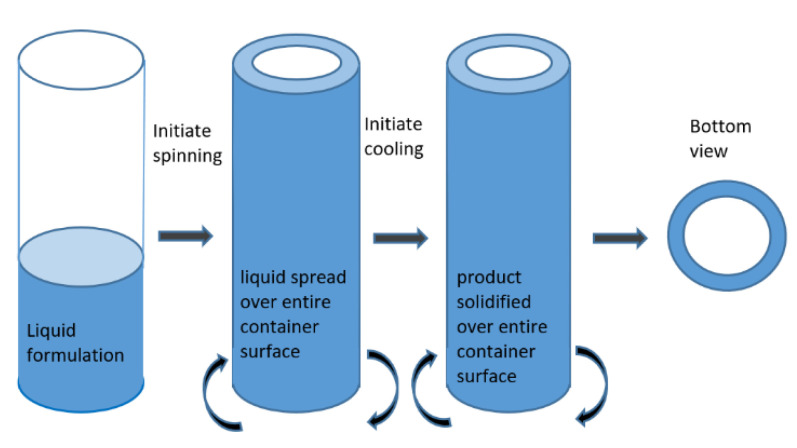
A method of increasing the evaporation surface and reducing the layer thickness in the spin freezing process; adopted from [35].

**Figure 3 foods-09-01488-f003:**
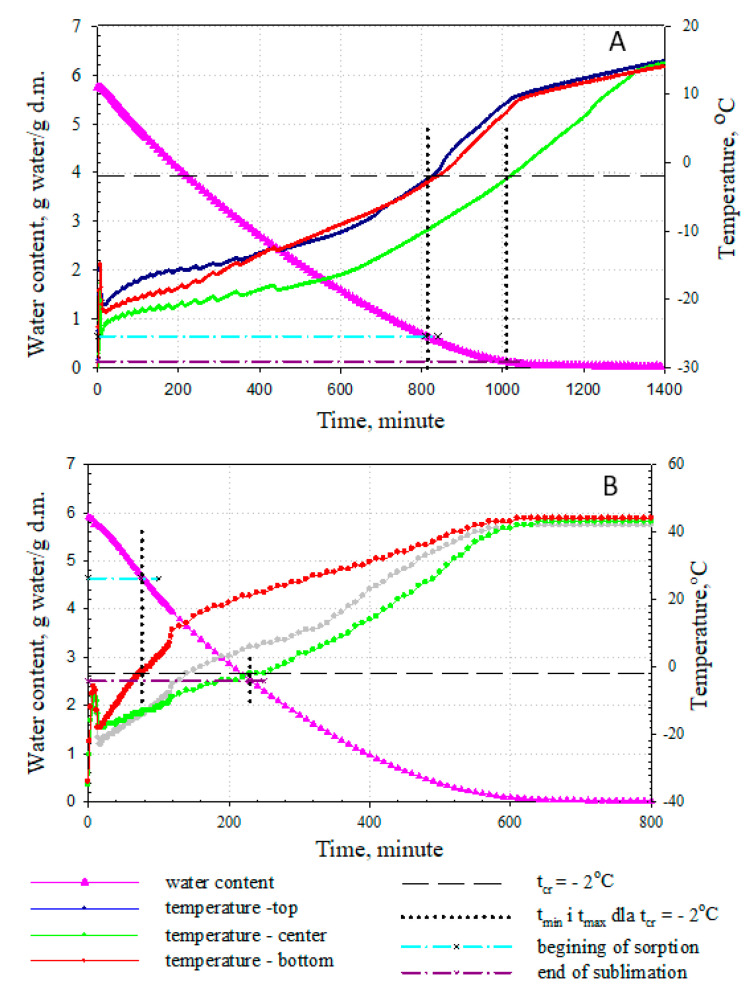
Kinetics of changes in the water content and material temperature measured along the slice axis and at its upper and lower surface during the freeze-drying of 1 cm-thick apple slices. The process was carried out at a pressure of 63 Pa and different shelf temperatures (T): (**A**)—T = 10 °C; (**B**)—T = 50 °C: adopted from [19].

**Table 1 foods-09-01488-t001:** Water activity and moisture content of selected freeze-dried products.

Material	Initial Moisture Content	Water Content After Drying	Water Activity of Freeze-Dried Product	Ref.
Grapefruit puree	83.0–86.7%	0.013–0.030 g water/g sample	n/a	[16]
Orange puree	n/a	<4%	n/a	[14]
Strawberry	90.8%	2%	n/a	[15]
Apple	86.3%	<0.5%	n/a	[15]
	n/a	n/a	0.14	[116]
Garlic	73.2%	0.061–0.095 g water/g sample	n/a	[94]
Arazá (*Eugenia stipitata* McVaugh) paste	96.0%	0.02 kg/kg d.m.	0.08	[117]
Pear	84.3%	<0.5%	n/a	[15]
Dragon fruit	86.5–87.5%	8.53–9.87%	0.08–0.16	[115]
Cooked rice	60.0%	1.69–2.09%	n/a	[118]
Hydrocolloid gels	83.6–87.5%	1.4–4.0%	0.14–0.330	[119]

n/a, not available.

**Table 2 foods-09-01488-t002:** Parameters of lyophilization and the properties of selected freeze-dried products.

Dried Material	Material Size and Form	Freezing Parameters	Shelf Temperature	Pressure of the Chamber	Drying Time	Properties of Material	Ref.
Abalone	Cylindrical disks (2.5 cm diameter and 0.7 cm height)	−40 °C	−45, −30, −20, −10, −5, 0, 10, and 15 °C	100 Pa	72 h	The increase of a shelf temperature in the range of −45–15 °C caused an increase of the apparent density of dried abalone from 372.9 to 472.1 kg/m^3^ and a decrease of apparent porosity from 0.733 to 0.664.	[93]
Apple	Cylindrical disks (2.5 cm diameter and 0.7 height)	−40 °C	−45, −30, −20, −10, −5, 0, 10, and 15 °C	100 Pa	72 h	The increase of a shelf temperature in the range of −45–15 °C caused an increase of apparent porosity from 0.876 to 0.910.	[93]
	Semi-circular slices (55 mm in length, 2.2–2.5 mm thick)	−25 °C	−25 °C at 1st step drying, 40 °C at 2nd drying	20 Pa at 1st step drying, 5 Pa at 2nd drying	24 h	The moisture content was negatively correlated with hardness of freeze-dried apples. The application of freeze-drying resulted in a lower springiness compared to that of the air-drying method. Additionally, the color difference ΔE of freeze-dried apples (11.37) was lower than that obtained for air-dried apples (21.11).	[116]
	Puree—layer with thickness of 4 mm	−40 °C	20 °C	63 Pa	26 h	The apple puree freeze-dried at 20 °C absorbed less water than air-dried samples. The application of freeze-drying method enabled the obtainment of powder with a slightly lower hygroscopicity than after microwave-drying.	[130]
Banana	Cylinders with diameter of 20 mm and 8 mm height	−35 °C (48 h), tempered for 1 h in liquid N_2_	Product temperature from −50 to −8 °C	3–300 Pa	24 h	The values of the bulk density of the banana decreased after freeze-drying from 1900 kg·m^−3^ to values lower than 400 kg·m^−3^. The values of bulk density increased (about ~30%) as the temperature of process was increased from −50 to −8 °C. The porosity of freeze-dried banana was the highest at the low temperature of −50 °C (~0.9).	[95]
Carrot	Cylinders with diameter of 20 mm and 8 mm height	−35 °C (48 h), tempered for 1 h in liquid N_2_	Product temperature from −50 to −5 °C	3–300 Pa	24 h	The values of bulk density of carrot tissue decreased after freeze-drying from 1750 kg·m^−3^ to values lower than 250 kg·m^−3^. The bulk density values increased (about ~40%) as the temperature of process was increased from −50 to −8 °C. The porosity of freeze-dried carrot was reduced by about 10% after drying at higher temperatures.	[95]
Coffee solutions	Layer with thickness of 20 mm	1 set: −40 °C at 1 °C/min 2 set: fluctuation of temperature between −40 and −20 °C	−40 °C at the primary drying, 20 °C at the secondary drying	10 Pa	18 h	Samples freeze-dried with temperature oscillations (−20 and −40 °C) had larger pores than material frozen at −40 °C. Temperature fluctuations during freezing promoted large crystal formation and resulted in a higher total porosity by, on average, 18%. The application of freezing cycles led to faster reconstruction rates.	[131]:
Dragon fruit	Pieces with thickness of 1 cm	−40 °C fast freezing (an air blast freezer and a contact plate freezer)	−5 °C at the primary drying, 30, 40, and 50 °C at the secondary drying	40 Pa	50 h at 30 °C, 55 h at 40 °C, 60 h at 50 °C	The apparent densities of freeze-dried dragon fruits were 0.16, 0.19, and 0.08 g × cm^−3^ at the drying temperatures 30, 40, and 50 °C, respectively. The hardness of dried fruit decreased from 9.26 to 4.33 N and crispness increased from 6.83 to 10.56 with the increase of the heating temperature.	[115]:
Eggplant	Cubes of 9 mm side	−40 °C	1 set: −30, −15, and 0 °C at 1st step drying, 20 °C at 2nd step drying 2 set: −30 and 0 °C at 1st drying, 20 °C at 2nd step drying	1 set: 10 Pa 2 set: 10, 20, and 40 Pa	1 set: 7–15.3 h 2 set: 14–20.9 h	The loss of antioxidant capacity was 49.9 and 68.6% for freeze-dried samples dried at −30 and 0 °C, respectively. The increase of drying temperature from −30 to 0 °C caused the loss of ascorbic acid from 37.9 to 12.2%. Total polyphenol content—TPC—in dried product was retained at higher pressures. The loss of TPC was 32.5% at 40 Pa and 47.7% at 10 Pa.	[132]:
Garlic	Brick shaped samples (20 × 10 × 10 mm)	−40 °C	−5, −15, and −25 °C	108 Pa	72 h	The decrease of shelf temperature from −5 to −25 °C during the freeze-drying of garlic resulted in a decrease of the apparent density from 469 to 431 kg/m^3^ and an increase of the shrinkage expansion between 0.44 and 0.52, as well as the true density decreased in the range of 1534–1504 kg/m^3^.	[94]:
Grapefruit puree	1-cm layer	−45 °C	room temperature, 40 °C	9 Pa	1.5−21 h	The increase of temperature promoted an increase in the porosity of freeze-dried puree (from 0.78 to 0.83) and a decrease in the number of pores formed from 415to 312.	[16]:
Gum Arabic solutions	Layer with a height of 0.5 cm	−40 °C (at 1 °C/min)	−20, −30, and −40 °C at primary during, 20 °C at secondary drying	10 Pa	18 h	The degree of puffing was stronger for samples dried at higher (−20 and −30 °C) compared to lower (−40 °C) temperatures of the shelf. The primary drying temperature did not affect the size of pores and pore distribution for solutions with concentrations of 20, 30, 40, and 50%. The mean pore diameter of 60% freeze-dried gum hydrocolloid system increased from 745 to 973 µm with the increase of shelf temperature from −40 to −20 °C.	[88]:
Kiwi	Whole fruit (without peel)	−40 °C	n/a	12, 20, 42, 85, and 103 Pa	n/a	The increase of pressure in the range of 12–100 Pa resulted in a decrease of *L** from 65.3 to 58.3, as well as *a** values from −2.7 to −6.8, and the increase of *b** from 22.3 to 28.3. The higher pressure affected the increase of penetration force for freeze-dried kiwi fruit from 4.3 to 16.2 N.	[121]:
Lime juice	Sample juice layer with a thickness from 0.3 to 1.1. cm	−30 °C	−61 °C	3 Pa	1–10 h	The freeze-drying of lime juice did not affect acidity (4.10–4.15 g citric ac./100 mL), antioxidant activity (17.5–18.3 mg ascorbic ac./100 mL), and carotenoids content (0.61–0.64 mg ∙100 mL^-^). Fresh juice and reconstituted freeze-dried juice did not significantly differ in relation to sensory attributes.	[133]:
Loco (*Concholepas concholepas*) (boiled)	Samples 1 × 1 × 0.5 cm Cubes 0.5 side cm	−25 °C	n/a	6,7 Pa and 9.6·10^−4^ Pa (AFD- atmospheric freeze-drying)	6.7−12 h	The pore surface of freeze-dried loco obtained at a low pressure was 0.32 m^2^ pores/m^2^, while after AFD, this value was half (0.16 pores/m^2^ material surface). The water absorption capacity of the freeze-dried sample was higher than 1.0 at a low pressure, while at AFD conditions, it was lower than 1.0.	[134]:
Maltodextrin sugar–agar solutions	Cube (10 × 10 × 10 mm) samples	−20, −40, and −80 °C, tempered at −80 °C before drying	Room temperature	10 Pa	48 h	The pore size and thickness of pore membranes of the freeze-dried system were reduced with a decrease of the freezing temperature. The system frozen at −80 °C was more resistant to compression than samples frozen at −40 and −20 °C.	[123]:
Orange puree	Puree, layer with a thickness of 0.5 mm	−45 °C—slow rate: a conventional freezer −38 °C—fast rate—a blast freezer	30, 40, and 50 °C	5 and 100 Pa	25 h at 30 °C, 7 h at 40 °C, 6 h at 50 °C	The color attributes *L*, *C**, and *h** of freeze-dried orange puree were affected by working pressure. The lower values of *L** and higher *C** were characteristic for samples dried at the high pressure of 100 Pa. The lower range of *h** values between 80.3 and 82.6 was registered for the samples dried at higher pressure (100 Pa) and the temperature of the shelf below 50 °C. The lower pressure of 5 Pa and a higher temperature of 50 °C created more resistant to fracture a freeze-dried sample. The lower degradation of vitamin C was observed for samples dried at 40 and 50 °C than at 30 °C.	[14]:
Pepper	Samples and puree with layer of 5 mm	−25 °C	20, 40, and 60 °C	63 Pa	290 min (60 °C) 900 min (20 °C)	The red pepper freeze-dried at higher temperature 60 °C was characterized by lower values of *L** (lightness =35.5), *a** (redness =27.6), and *b** (yellowness =23.8) than the sample dried at 20 °C (*L** = 39.2, *a** = 34.8, and *b** = 27.0). Additionally, the increase in drying temperature caused a decrease of the total phenolic content (from 12.6 to 11.8 mg GAE/g d.m.) and antioxidant activity (EC50- concentration required to obtain a 50% antioxidant effect) from 21.7 to 26.1 mg d.m./mL).	[11]:
Potato	Cylindrical disks (2.5 cm diameter and 0.7 height)	−40 °C	−45, −30, −20, −10, −5, 0, 10, and 15 °C	100 Pa	72 h	The increase of a shelf temperature in the range from −45–15 °C caused an increase of apparent density of dried potato from 204.2 to 452.2 kg × m^−3^ and a decrease of apparent porosity from 0.863 to 0.698.	[93]
Rice (cooked)	The layer of 1.8 mm	−18 °C	90 °C	80 Pa (initial) and 20 Pa (final)	12 h	The freeze-dried rice had a better rehydration capacity than the freshly cooked sample. Freeze-drying caused the extensive breakage of the grains. The extent of breakage was dependent on the cooking method and was lower in freeze-dried parboiled rice (3.6–36.9%) than non-parboiled grain (50%).	[118]:
	n/a	−30 °C for 72 h, tempered for 1 h in liquid N_2_	n/a	4, 13, and 125 Pa	24 h	The bulk density of freeze-dried rice decreased from ~0.9 to ~0.8 for kernels boiled for 4 min and from ~0.6 to ~0.5 for kernels boiled for longer than 20 min with the decrease of applied pressure from 125 to 4 Pa. The porosity of dried kernels was the highest at low pressures.	[10]:
Strawberry	Whole fruits and slices (5 and 10 mm thick)	−40 °C	30, 40, 50, 60, and 70 °C	n/a	12–48 h	The color of strawberries and the volume reduction of fruits did not change in case of different drying temperatures. The percentage of collapsed samples exceeded 20% at drying temperatures higher than 50 °C.	[99]:
Yellow dates	Halves	−40 °C	−45, −30, −20, −10, −5, 0, 10, and 15 °C	100 Pa	72 h	The increase of a shelf temperature in the range from −45–15 °C caused a decrease of the apparent density of dried dates from 485.1.2 to 205.5 kg·× m^−3^, as well as an increase of apparent porosity from 0.709 to 0.864.	[93]:

n/a, not available.

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
