# Peer review of "The Freeze-Drying of Foods—The Characteristic of the Process Course and the Effect of Its Parameters on the Physical Properties of Food Materials"

_foods, 2020, doi:10.3390/foods9101488_

Round 1

Reviewer 1 Report

In this review, the authors analyze in detail the freeze-drying literature. In particular, they emphasize the various stages of the process and the effects of the process conditions on the quality of the final product. It is a good article. However, some minor changes could improve it. 

Specifically:

1. The first part of the introduction is identical to the abstract: it needs to be changed.

2. Many lists are not numbered (eg lines 58-64 or even 97- 108). Numbering increases readability.

The quality of figure 3 should be improved.

Reviewer 2 Report

The manuscript foods-933983-peer-review-v1 “The freeze-drying of foods – the characteristic of the process course and the effect of its parameters on the physical properties of food materials” contains interesting information on freeze-drying process application to foods. To my opinion, the authors extend very much in the description when report the characteristics of the freeze-drying process moreover some repetitions are observed in the manuscript. Exists recent literature that deals with fundamentals of freeze-drying (Assegehegn et al (2020) Freeze-drying: A relevant unit operation in the manufacture of foods, nutritional products, and pharmaceuticals. Adv. Food Nutr. Res. 93:1-58). The authors must stress how this method should be correctly used for food products in order to obtain a high-quality freeze-dried product at low cost. Several points need very careful revision. Suggestions about the manuscript are listed following.

  • Introduction: This section needs to be rewritten in order to make it more clear to the reader.
  • Lines 32-34: Use bullets to report the steps of the freeze-drying process.
  • Line 39: Remove "another method of"
  • Lines 44-46: "The effect of freeze-drying conditions....." you have to move this sentence in the line 48 before the last paragraph. Report briefly what is the main effect on nutritional properties antioxidants etc and link it with the aim of this review. E.g although there is considered the best drying method has some drawbacks since is reported to negatively affect the colour …browning ….of the foods.
  • Lines 47-48: Sentence "Therefore, the characteristic ........" is not clear what do you want to say. Rephrase.
  • Line 57: Use bullets to report the six steps
  • Line 64: "The parameters in which these phenomena occur determine" - rephrase the sentence.
  • Lines 82-83: "However, it does not mean that the process runs at the maximum sublimation rate possible under given conditions" - Why ... because of what? Explain.
  • Line 97: Use bullets.
  • Lines 111-143: The text in these lines is not clearly organized. Here a general discussion about the freezing rate should precede the first paragraph that should be linked to the paragraph that deals with the case that is needed to preserve the biological functions.
  • Following, the discussion could be organized about the effect on different matrixes (materials). Moreover, you can add table reporting the freezing type and the temperature rates advisable to different substrates.
  • Line 118: Since "frozen cake" could lead to misleading explain what do you mean by cake next in parenthesis.
  • Line 134. The paragraph beginning with “The above-mentioned results concerned the lyophilisation of the solution or the pulp" should be in the same paragraph with the previous one since are explaining the same thing.
  • Lines 203-207: Sentence "When designing the lyophilisation ..." needs rephrasing.
  • Line 282: "In the pharmaceutical industry, ....." add in parenthesis types of materials used.
  • Lines 298-300: Rephrase the sentence.
  • Line 300: 7.9 10-4 kg ....m2 use superscript.
  • Figure 3. Move figure 3 after the text citing it. Moreover, provide a more legible figure.
  • In section 3: “Effect of freeze-drying conditions of the selected physical properties of materials” there are observed repetitions of what authods say in the previous section. Since they explain in detail in the previous section the theoretical background of the process, they must focus only on the application of freeze-drying process on the food products (stressing how the food properties are affected) that represent the aim of this review.
  • Table 1 : Report the table after the text citing it. Moreover, the table must have another column to report the initial water content/aw of these materials.
  • Table 2 : The table was not reported in the text. Check the page numbering for the table as well as the table name in line 725 (you name the same table Table 1 cont.). You have to remove the header line and link this with the previous part of the table 2. The table must have another column to report the effect on the food properties observed. In addition, the table is organized based on the cited literature. Better organize it based on the type of food or initial aw.
  • Section 4 named "4. Discussion and Conclusions " should be reported as "Conclusions". This section should be shortened and focus on new directions of possible further steps to reduce the negative effects of this process in foods.

Reviewer 3 Report

Foods-933983

 The freeze-drying of foods – the characteristic of the 2 process course and the effect of its parameters on the 3 physical properties of food materials 4

Dorota Nowak, Ewa Jakubczyk*

This is a review paper on freeze-drying of food, focused on influence of operating parameters on food properties. The quality of the paper is sufficient to recommend revision, but some improvements are required.

Major point:

Two main aspects are investigated: the effect of operating conditions on food physical properties, and on activity of the product. This second aspect is common to freeze-drying of pharmaceutical products, as clearly discussed in the conclusions section: literature on this is relatively small for foods, much more abundant on pharmaceuticals, and it is reasonable that the authors refer to the latter literature. But in some cases the discussion (in the first part of the paper) refers to the pharma literature without taking into account the important differences; for example results in case of freeze-drying of liquids in vials are hardly applicable to food products. Some specific examples are listed below.

Below some specific comments are added:

  • Line 73-74, 84-86. The authors refer to “monitoring” and “control”.

One way to monitor the equilibrium state is to control the temperature of the lyophilized  material [19].

On the other hand, an increase in temperature may mean too much heat input, but may also be the effect of reduced evaporation due to a low water content. Therefore, for more complete control, changes in  the water content should also be monitored simultaneously [19]

It is not clear this part, and what the authors mean by “monitor equilibrium state”. Do the author refer to real monitoring? They refer to a publication not easily available, and it would be better to add some details. In any case there is a wide literature (also some recent review papers on monitoring and control in freeze drying) and some additional references would be appropriate.

In any case I would not say that increase in T is due to reduced evaporation due to a low water content. It is always a consequence of too much heat supply, may be with respect to the possible consumption by sublimation, for increased resistance.

  • Line 97: The main purposes of freezing the product before freeze-drying are…

The product must always be frozen. May be the authors mean freezing before loading in the freeze-dryer?

  • Line 144-48: SFD. The FD of granular material has advantages and disadvantages that should be discussed (the reduced heat conduction in the granular bed negatively affects the process). Two recent reviews on SFD should be mentioned. [Wanning et al., International Journal of Pharmaceutics 488 (2015) 136–153; Adali et al., Processes 2020, 8, 709; doi:10.3390/pr8060709]

  • Line 157-62: spin feezing. This is an example of a technology proposed for pharma; it can be applied to FD in vials, I cannot see how this concept can be usefully employed for food. At least the authors should comment.

  • Line 178: evaporative freezing. Other older references may be relevant. See e.g Elia A.M., Barresi A.A., 1998, Intensification of transfer fluxes and control of product properties in freeze-drying. Eng. Process. 37(5), 347-358

-        Line 287-91: This part describes heat transfer mechanisms, but reference and discussion if for products in vials. In case of food may be it is more interesting to discuss mechanisms for heat transfer in tray, where the point-contact is the most relevant mechanism. Significant literature is available (I remember a paper by Bruttini, The Chemical Engineering Journal Volume 45, Issue 3, February 1991, Pages 175-177 as the first, but other literature is available)

  • Line 352-55. May be it would be more significant to report the residual moisture at end of primary drying as % of the dry material, which is typical of the product. These data are commonly reported in books for main excipients.

  • Figure 3. Too small, text is not readable.

  • Line 461: When the weight of the material does not change within 30 minutes, it can be assumed that secondary drying is complete.

Weighing material is possible, in lab, but hard in industrial plants. This concept should be better detailed. In any case weighing is suitable for end of primary drying, very difficult to determine end of secondary drying. There is some literature on weighing samples in FD, of which the authors do not seem to be aware.

  • Line 526-7: The smaller pore size obtained at a lower shelf temperature was explained by the collapse of the structure due to slower sublimation

This seems wrong to me, and in any case should be better explained. Pore size depends mainly on freezing; here it seems that the authors refer to reduction of pore size due to collapse, but collapse is not caused by slower sublimation!. And in any case, it seems strange that collapse is favoured by lower shelf temperature!

  • Line 611: product temperature, moisture, deformation are not process parameter (process parameters may be shelf temperature and chamber pressure); eventually state variable.
  •  
  • Line 628-30: The minimal texture degradation of freeze-dried strawberries was observed at a shelf temperature of 15 and 45℃; however, the rise of temperature should be  maintained at 1.6 and 0.4℃/min for the lower and higher plate temperatures, respectively [110].

This part is criptic, and should be better clarified. It is not clear what temperature rise refers to, and why is relevant.

  • The authors should take care of using a uniform style; actually reference list is a mess, some title in uppercase, other with Upper case initial or not, journal name abbreviated or not,….

Typos:

Line 11: “high” duplicated

Line 144: it is possible to obtain porosity of the material?

Line 284: The heat flux is the greater

In all the manuscript pedex and apex are misprinted

Caption of figure 3.

Line 459-60. Reference to Figure 1 seems wrong

Line 713: ujarit?

Round 2

Reviewer 2 Report

The manuscript foods-933983 “The freeze-drying of foods – the characteristic of the process course and the effect of its parameters on the physical properties of food materials” in its revised version is improved. Nevertheless, there is needed a final editing . e.g. line 319 remove "r" and one dot.

Author Response

The manuscript foods-933983 “The freeze-drying of foods – the characteristic of the process course and the effect of its parameters on the physical properties of food materials” in its revised version is improved. Nevertheless, there is needed a final editing . e.g. line 319 remove "r" and one dot.

Response: The correction was made in the paper.

Reviewer 3 Report

Manuscript ID: foods-933983 - Revised

Minor revison required.

  1. If we dry e.g. apple slices with peel, plum slices or vegetable pulp in molds, the side surface is insulated by the peel or mold. Thus, the heat and mass exchange occur also only in one direction. So the model of heat transport is similar

This is not convincing, it is hard to believe that peel may insulate; it can strongly reduce mass transfer, but not heat transfer.

  1. The smaller pore size obtained at a lower shelf

597 temperature was explained by the collapse of the structure

The authors have not explained in a convincing way, and I do not remember to have ever found in literature collapse occurring reducing shelf temperature.

It may be eventually caused by increased temperature when residual moisture is still high (as a consequence of slower drying). Is this the cae? Otherwise better to delete that part, if nor explained in a reasonable way.

  1. Reaching the cryoscopic temperature when the lyophilised material has a high water content, 509 higher than the bound water content, indicates the melting of the ice (Fig 1 A), while when it is 510 reached with a sufficiently low water content, the process was correct (Fig 1 B).

Wrong figure number

  1. References: I noticed that new references have been inserted, but not added, substituted to other. Is this correct? Or it was a trick to avoid renumbering? Suggestion was to add missing references, not to delete some of the exixting one, that seemed to be
